# Ouabain Counteracts Retinal Ganglion Cell Death Through Modulation of BDNF and IL-1 Signaling Pathways

**DOI:** 10.3390/brainsci15020123

**Published:** 2025-01-26

**Authors:** Amanda Candida da Rocha Oliveira, Camila Saggioro Figueiredo, Ícaro Raony, Juliana Salles Von-Held-Ventura, Marcelo Gomes Granja, Thalita Mázala-de-Oliveira, Vinícius Henrique Pedrosa-Soares, Aline Araujo dos Santos, Elizabeth Giestal-de-Araujo

**Affiliations:** 1Department of Neurobiology and Program of Neurosciences, Institute of Biology, Federal Fluminense University, Niteroi 24020-141, Brazil; amanda.oliveira2909@gmail.com (A.C.d.R.O.); camila.saggioro@outlook.com (C.S.F.); julianasvhventura@gmail.com (J.S.V.-H.-V.); thalitamazala@id.uff.br (T.M.-d.-O.); viniciushen@gmail.com (V.H.P.-S.); 2National Institute of Science and Technology on Neuroimmunomodulation—INCT-NIM, Oswaldo Cruz Foundation, Rio de Janeiro 21041-250, Brazil; alinerabelo@id.uff.br; 3Institute of Medical Biochemistry Leopoldo de Meis, Federal University of Rio de Janeiro, Rio de Janeiro 21941-902, Brazil; icraony@gmail.com; 4Research, Innovation, and Surveillance Center for COVID-19 and Health Emergencies, Oswaldo Cruz Foundation, Rio de Janeiro 21040-360, Brazil; marceloggranja@gmail.com; 5Department of Physiology and Pharmacology, Biomedical Institute, Federal Fluminense University, Niteroi 24210-130, Brazil

**Keywords:** brain-derived neurotrophic factor, cytokine, growth factor, steroid hormone, interleukin-1β, matrix metalloproteinase-9, ouabain, neuroprotection, retinal ganglion cell, tumor necrosis factor

## Abstract

**Background:** Ouabain is a steroid hormone that binds to the sodium pump (Na^+^, K^+^-ATPase) at physiological (nanomolar) concentrations, activating different signaling pathways. This interaction has been shown to prevent the axotomy-induced death of retinal ganglion cells (RGCs), although the underlying mechanisms remain unclear. **Objective:** In this study, we investigated potential mechanisms by which ouabain promotes RGC survival using primary cultures of rat neural retina. **Results:** Our findings indicate that ouabain regulates brain-derived neurotrophic factor (BDNF) signaling in retinal cells via matrix metalloproteinase-9-mediated processing of proBDNF to mature BDNF (mBDNF) and by increasing the phosphorylation of the mBDNF receptor, tropomyosin-related receptor kinase B. Ouabain also enhances the maturation of interleukin (IL)-1β through the increased activation of caspase-1, which mediates the processing of proIL-1β into IL-1β, and transiently upregulates both IL-1 receptor and IL-1 receptor antagonist (IL-1Ra). Treatment using either IL-1β or IL-1Ra alone is sufficient to enhance RGC survival similarly to that achieved with ouabain. Finally, we further show that ouabain prevents RGC death through a complex signaling mechanism shared by BDNF and IL-1β, which includes the activation of the Src and protein kinase C pathways. **Conclusions:** Collectively, these results suggest that ouabain stimulates the maturation and signaling of both BDNF and IL-1β, which act as key mediators of RGC survival.

## 1. Introduction

Retinal ganglion cells (RGCs) are specialized projection neurons that convey visual information from the retina to the superior colliculus and the dorsal lateral geniculate nucleus of the thalamus. Over recent decades, numerous studies using in vitro or in vivo models have identified various factors that either inhibit or promote RGC survival, such as neurotransmitters, cytokines, growth factors, and hormones [1,2,3]. This is particularly important since RGC death can severely impact vision and is the leading cause of irreversible blindness worldwide [4]. Among the factors implicated in RGC survival is ouabain, a steroid hormone that, at physiological (nanomolar) concentrations, binds to the sodium pump (Na^+^, K^+^-ATPase, NKA) and prevents axotomy-induced RGC death through the subsequent activation of the Src kinase, epidermal growth factor receptor (EGFR), protein kinase C (PKC)-δ, and c-Jun N-terminal kinase (JNK) pathways [5,6].

Interestingly, ouabain has been shown to exert a synergistic effect with brain-derived neurotrophic factor (BDNF) [7], a growth factor known for its trophic effect on RGCs [8]. This synergism occurs through the activation of the Src, PKC-δ, and JNK pathways [7]. However, for RGC survival, ouabain also requires the activation of nuclear factor-kappa B (NF-κB) [7], which induces the expression of genes encoding proinflammatory cytokines, such as interleukin-1 beta (IL-1β) and tumor necrosis factor-alpha (TNF-α) [9]. Although IL-1β and TNF-α are classically recognized as proinflammatory cytokines, at low concentrations, both can promote RGC survival and axon regeneration without detrimental effects [10,11]. Accordingly, ouabain induces low-level secretion of these cytokines by retinal cells and blocking either IL-1β or TNF-α abrogates ouabain-induced RGC survival [11].

In retinal cells, ouabain also engages matrix metalloproteinase-9 (MMP-9) to cleave and subsequently release TNF-α [11,12]. Furthermore, ouabain modulates TNF receptor levels, decreasing TNF receptor 1 (TNFR1), which mediates the proinflammatory and neurodegenerative effects of TNF-α, while increasing TNF receptor 2 (TNFR2), which is associated with pro-survival and neuroprotective effects [12]. This evidence led us to hypothesize that ouabain may regulate the synthesis and signaling of cytokines and growth factors in retinal cells, thereby establishing a functional redundancy mechanism to promote RGC survival. Here, using primary cultures from rat neural retina, we further show that ouabain regulates the maturation of BDNF and IL-1β and modulates the levels of their respective receptors in retinal cells. We also provide evidence that ouabain, BDNF, IL-1β, and TNF-α can promote RGC survival through the activation of redundant signaling pathways.

## 2. Materials and Methods

### 2.1. Materials

Recombinant IL-1β and TNF-α were purchased from PeproTech (Cranbury, NJ, USA). Fetal calf serum (FCS), medium 199, and trypsin were purchased from Gibco (Gaithersburg, MD, USA). JNK inhibitor and MMP-9 inhibitor were sourced from Calbiochem (San Diego, CA, USA). Petri dishes came from TPP (Trasadingen, Switzerland). Ouabain, Penicillin G, streptomycin sulfate, L-glutamine, poly L-ornithine, horseradish peroxidase (HRP), tetramethylbenzidine, sodium nitroprusside, dimethylsulfoxide (DMSO), sodium dodecyl sulfate (SDS), Tween 20, AG-1478, chelerythrine chloride, 4-amino-5-(4-methylphenyl)−7-(t-butyl)pyrazolo[3,4-d]pyrimidine (PP1), and rottlerin were obtained from Sigma-Aldrich (St. Louis, MO, USA). Luminata Forte HRP was bought from Millipore (Burlington, MA, USA). Bovine serum albumin (BSA) was bought from Fisher Scientific (Waltham, MA, USA). Ponceau red, hydrogen peroxide, glycerol, 2-mercaptoethanol, and xylene were purchased from VETEC Química Fina (Duque de Caxias, RJ, Brazil). Nitrocellulose membranes were sourced from GE Healthcare Life Sciences (Marlborough, MA, USA). Paraformaldehyde, glutaraldehyde, glycine, methanol, and sodium phosphate dibasic and monobasic were supplied by J.T.Baker (Phillipsburg, NJ, USA). Entellan was supplied by Merck (Darmstadt, Germany). Tris (hydroxymethyl) aminomethane was obtained from Êxodo Científica (Sumaré, SP, Brazil). Acrylamide, methylene bis-acrylamide, and tetramethylethylenediamine were sourced from Amersham (Piscataway, NJ, USA).

### 2.2. Animals

This study was conducted using male and female Lister Hooded rats, sourced from the Animal Facility at Fluminense Federal University. The animals were housed under standardized conditions, including a 12 h light/dark cycle and a controlled ambient temperature of 22–25 °C, with unrestricted access to water and food. All experimental procedures adhered to the guidelines outlined in the National Institutes of Health’s Guide for the Care and Use of Laboratory Animals and were approved by the Ethical Committee of Fluminense Federal University (protocol #8987300921). Furthermore, every effort was made to minimize the number of animals used and to reduce any potential distress or discomfort.

### 2.3. Primary Retinal Cell Cultures

Primary retinal cultures were prepared as previously described [12,13]. Newborn rats (P2) were euthanized, and their neural retinas were extracted in CMF solution with antibiotics (100 μg/mL streptomycin and 100 U/mL penicillin). The retinas were then incubated with 0.1% trypsin for 18 min at 37 °C, after which the trypsin was neutralized using culture medium (199 medium supplemented with 2 mM glutamine, 100 μg/mL streptomycin, and 100 U/mL penicillin) enriched with 5% FCS. Retinal cells were mechanically dissociated using a Pasteur pipette, plated in poly-L-ornithine-coated Petri dishes at a density of 10^5^ cells/cm^2^ and incubated in culture medium for 2 h to allow cell attachment. Then, cultures were treated once with ouabain (3 nM), recombinant IL-1β (5 ng/mL), recombinant TNF-α (0.5 ng/mL), or recombinant IL-1 receptor-antagonist (IL-1Ra; 25–200 ng/mL). For some experiments, retinal cultures were simultaneously exposed to the following drugs: MMP-9 inhibitor (20 nM), a Src inhibitor PP1 (1 µM), PKC inhibitor chelerythrine chloride (CC, 1.25 µM), PKCδ inhibitor rottlerin (2 µM), EGFR inhibitor AG-1478 (2.5 µM), and JNK inhibitor (1 µM). Drug concentrations were based on previously published reports [5,6,7,11,12,13]. All cultures were maintained in a humidified atmosphere of 5% CO2/95% air at 37 °C, and each experiment was performed with at least three independent cultures, with different rat litters being used for each culture. The experimental timeline is illustrated in Figure 1a.

### 2.4. Retrograde Labeling, Identification, and Quantification of Retinal Ganglion Cells

Retrograde labeling of RGCs was performed following an established protocol [12,13] (Figure 1b). For this, newborn rats (P1) were randomly selected, anesthetized by hypothermia, and injected with a solution of 30% HRP diluted in 2% DMSO. A 1 µL volume of this solution was administered directly into each superior colliculus using a Hamilton syringe. The soft skull and thin skin of the newborn rats facilitated accurate injections. The animals were returned to their mothers, and after 16 h, the HRP had been retrogradely transported to the RGC soma. Cultures were then prepared for analysis as described above (see Section 2.4). HRP in the RGC soma was visualized following Mesulam’s staining protocol [14]. As previously described [12,13], cultures were fixed with Karnovsky’s solution and cells were reacted with tetramethylbenzidine and hydrogen peroxide. The coverslips were washed, air-dried, and mounted on entellan before manual counting of labeled RGCs under a microscope Olympus BX41 (Tokyo, Japan). Three distinct and continuous areas of the coverslip were randomly chosen for analysis. To control variability in HRP labeling across experiments, cell counts at 4 h in culture were considered 100%, and all data were presented as percentages of this baseline. In addition, a double-blind process was implemented to minimize the researcher’s influence on the experimental results. Consistently, around 50% of HRP-labeled RGCs died in control conditions after 48 h in culture in comparison to 4 h (Appendix A).

### 2.5. Western Blotting

Western blotting procedures were carried out following protocols previously reported [12,13]. In summary, retinal cell cultures and retinal tissue from rats at the postnatal stages P0, P2, P7, P14, and P30 were lysed by homogenization in lysis buffer. Lysates (60 µg of total protein per lane) were separated on 9–15% SDS-polyacrylamide gel electrophoresis (SDS-PAGE) gels, and the proteins were transferred to nitrocellulose membranes using a semi-dry transfer system (Bio-Rad Laboratories, Hercules, CA, USA). The membranes were blocked with 3% (*w*/*v*) bovine serum albumin (BSA) in TBS-T buffer (Tris-buffered saline pH 7.4 plus 20 mM Tris–HCl, 160 mM NaCl, and 0.1% Tween 20) for 1 h at room temperature. Membranes were incubated overnight at 4 °C with primary antibodies (see Table 1) diluted in the blocking solution. Afterward, the membranes were treated with secondary antibodies (see Table 1) for 1 h at room temperature. Detection was performed using the Luminata reagent (Millipore, Burlington, USA), and images were captured with a ChemiDoc-Pix System (Loccus Biotecnologia^®^, Cotia, Brazil). Quantification of optical density was performed using ImageJ software (version 1.54; NIH, Bethesda, MD, USA), with β-actin as the loading control.

### 2.6. Statistical Analysis

Statistical analyses were conducted using GraphPad Prism 10 software. All data exhibited a normal distribution, as assessed by the Shapiro–Wilk test. Comparisons between two groups were analyzed by two-tailed Student’s *t*-test. Comparisons between multiple experimental groups were analyzed using one-way analysis of variance (ANOVA). ANOVA revealing a significant effect (*p* < 0.05) was followed by post hoc Holm–Sidak’s test. *p*-value < 0.05 was considered statistically significant. All data are expressed as mean ± standard error of the mean (SEM).

## 3. Results

### 3.1. Ouabain Stimulates Maturation of BDNF and TrkB Phosphorylation

BDNF was the second member of the neurotrophin family to be identified for its trophic effects on sensory neurons [15]. Other members of this family also include nerve growth factor (NGF), neurotrophin 3, and neurotrophin 4/5 [16]. These molecules share several similarities, including their sequence, structure, and processing homologies. Synthesized initially as pro-neurotrophins (immature precursors), they undergo proteolytic cleavage to form mature proteins [16]. To determine ouabain’s effect on the maturation of BDNF, we assessed the protein levels of proBDNF and mature BDNF (mBDNF) in retinal cells after exposure to ouabain for different periods. We found that ouabain promoted an increase in the pro-BDNF levels concomitant with a decrease in the mBDNF levels after 15 min and 24 h (Figure 2a,b). However, after 48 h, the retinal cell cultures exposed to ouabain showed a reduction in the proBDNF levels associated with a significant increase in the mBDNF levels (Figure 2c). No significant changes were observed in the levels of mature NGF (Appendix A), the first identified growth factor [17].

While mBDNF has a high affinity for tropomyosin-related kinase B receptor (TrkB), a receptor well known for its role in cell survival and growth, proBDNF primarily binds to p75 neurotrophin receptor (p75NTR), typically triggering cell apoptosis [18]. Therefore, we also evaluated whether ouabain modulates the levels of p75NTR and phosphorylated TrkB (pTrkB). Exposure to ouabain did not significantly change the p75NTR levels but increased the pTrkB levels after 48 h (Figure 2d and Appendix A). Taken together, these findings suggest that ouabain stimulates the synthesis, subsequent maturation, and signaling of BDNF.

### 3.2. MMP-9 Mediates Ouabain-Induced Maturation of BDNF

MMP-9 is a member of a large family of zinc-dependent endopeptidases and acts to cleave components of the extracellular matrix as well as cell adhesion molecules, cell surface receptors, cytokines, and growth factors [19]. Accordingly, evidence has indicated that MMP-9 also plays a critical role in the maturation of BDNF by processing proBDNF to mBDNF [20,21]. To test the impact of MMP-9 activity on the ouabain-induced maturation of BDNF, we treated retinal cells with an MMP-9 inhibitor (MMP-9i) and ouabain. Treatment with MMP-9i for 48 h inhibited an ouabain-induced proBDNF decrease and mBDNF increase (Figure 3a–c), suggesting that ouabain depends on MMP-9 activity to stimulate the maturation of BDNF.

### 3.3. Ouabain Promotes a Physiological IL-1β Signaling

IL-1β is also synthesized as an immature precursor (proIL-1β) and must be post-translationally processed by active caspase-1 to generate IL-1β [22]. Once released, IL-1β binds to its cognate receptor IL-1R1 and recruits the IL-1 receptor accessory protein (IL-1RAcP) to initiate signaling [22]. Previously, we showed that treatment with ouabain for 15 min increases the IL-1β levels in both the lysate and supernatant of retinal cell cultures [11], suggesting that ouabain stimulates, respectively, the synthesis and release of IL-1β by retinal cells. To gain insight into the mechanisms by which ouabain regulates IL-1β signaling in retinal cells, we assessed the protein levels of active caspase-1 and IL-1R1 after exposure to ouabain for different periods. Our results indicated that treatment with ouabain transiently increased the levels of both active caspase-1 and IL-1R1 after 15 min (Figure 4a–f).

Next, we also evaluated whether ouabain modulates the levels of IL-1R antagonist (IL-1Ra). This cytokine also binds to IL-1R1 but does not activate the receptor, thus acting as a competitive antagonist of the IL-1β signaling pathway [22]. We found that ouabain decreased IL-1Ra in the cell lysates after 15 min, followed by an increase after 48 h (Figure 5a–c), which may be a mechanism by which ouabain prevents aberrant IL-1β signaling. Accordingly, treatment with low concentrations (50–150 ng/mL) of IL-1Ra was sufficient to increase RGC survival (Figure 5d), similar to ouabain and IL-1β [11]. Taken together, these results suggest that ouabain stimulates IL-1β signaling by promoting the caspase-1-mediated maturation of IL-1β, subsequent IL-1β release, and IL-1R1 upregulation, while it also engages IL-1Ra to ensure non-aberrant IL-1β signaling and RGC survival.

### 3.4. Ouabain Promotes RGC Survival Through a Signaling Mechanism Shared by IL-1β, TNF-α, and BDNF

IL-1β and TNF-α not only mediate the trophic effect of ouabain but they are also sufficient to promote RGC survival [11]. However, the signaling pathways underlying IL-1β- and TNF-α-induced RGC survival remain to be determined. Since the effect of ouabain on RGC survival involves the activation of the Src, EGFR, PKC-δ, and JNK pathways [5,6], we treated retinal cell cultures with pharmacological inhibitors of these signaling pathways in the presence of IL-1β or TNF-α and assessed the number of HRP-labeled RGC after 48 h (Figure 6a–d and Appendix A). Our results indicated that treatment with PP1 (Src inhibitor, Figure 6a), AG-1478 (EGFR inhibitor, Figure 6b), and chelerythrine chloride (pan-PKC inhibitor, Figure 6c) prevented IL-1β-induced RGC survival. However, treatment with a rottlerin (PKC-δ inhibitor, Figure 6c) or a JNK inhibitor (JNKi, Figure 6d) was not able to inhibit the trophic effect of IL-1β. By comparison, treatment with AG-1478 and chelerythrine chloride abolished TNF-α-induced RGC survival, whereas PP1 did not (Appendix A). Table 2 summarizes the signaling pathways underlying the trophic effect induced by ouabain, IL-1β, TNF-α, and BDNF. Overall, these findings suggest that the Src, EGFR, and PKC pathways represent a common mechanism of regulating RGC survival shared by ouabain, IL-1β, TNF-α, and BDNF.

## 4. Discussion

Vision loss in retinal diseases predominantly results from the degeneration and death of RGCs. In the present study, we used axotomy-induced RGC death as a model of injury to study molecules capable of promoting neuronal survival. Here, we demonstrate that ouabain plays a significant role in fostering RGC survival by modulating multiple neurotrophic and inflammatory pathways. Specifically, ouabain enhances BDNF maturation through the stimulation of MMP-9, accompanied by a decrease in the proBDNF levels and increased phosphorylation of TrkB receptors in 48 h, which supports the synergistic effect between ouabain and BDNF leading to RGC survival [7]. Additionally, our findings show that ouabain induces IL-1β signaling through caspase-1 activation and IL-1R1 upregulation while maintaining physiological control over IL-1β signaling via IL-1Ra modulation. Moreover, our study shows that both IL-1β and TNF-α promote RGC survival via EGFR and PKC pathway activation.

In the visual system, BDNF has long been established as a critical survival factor for RGCs following various types of axonal injury [23,24,25,26]. For instance, studies in animal models, such as rats, have demonstrated that intravitreal injection of BDNF can preserve RGC viability for up to 1 week following axotomy, whereas untreated axotomized RGCs exhibited nearly 50% cell death [27]. Additionally, BDNF treatment was shown to delay axotomy-induced RGC degeneration by 2–3 fold after 3 weeks [28]. Abnormalities in BDNF signaling have been observed across multiple models of retinal diseases [29,30], including glaucoma [31], and there is accumulating evidence that diminished BDNF and TrkB signaling may also play a role in the pathology of human glaucoma [32,33]. RGCs express both BDNF receptors, TrkB and p75NTR, which mediate different cellular responses depending on the form of BDNF present [34]. Consequently, insufficient BDNF processing can lead to the accumulation of proBDNF, which preferentially binds to the p75NTR in retinal glial cells, potentially resulting in its activation. This activation has been associated with RGC loss, as it shifts the signaling balance toward apoptotic pathways rather than cell survival [35].

Our current findings, complemented by our previous results [7], indicate that ouabain likely shifts the neurotrophic signaling balance toward survival pathways by reducing the proBDNF levels and increasing mBDNF within 48 h. This shift counteracts apoptotic signaling mechanisms, thereby promoting neuronal survival, as previously described. Moreover, the increase in mBDNF and phosphorylated TrkB levels further supports TrkB-mediated pro-survival signaling, a crucial mechanism given that TrkB mRNA levels in RGCs decline shortly after optic nerve injury [36]. Notably, combining TrkB gene delivery to RGCs with exogenous BDNF has been demonstrated to significantly extend RGC survival following axonal injury [36], suggesting that strategies aimed at promoting TrkB expression could enhance the neuroprotective effects of BDNF. In this context, ouabain’s capacity to boost mBDNF and stimulate TrkB phosphorylation may offer a promising approach to preserve RGCs.

Additionally, our findings demonstrate that MMP-9 plays a crucial role in BDNF maturation induced by ouabain after 48 h, highlighting its potential impact on retinal cell survival. Consistent with previous studies showing that MMP-9 promotes the conversion of proBDNF into mBDNF in human cell lines and mouse brain cells [20,21], our results also suggest that a similar process occurs in retinal cells following ouabain treatment. However, the precise mechanisms by which ouabain triggers MMP-9-dependent BDNF maturation remain to be elucidated, necessitating further investigation.

Over the past decade, ouabain has gained recognition for its potent immunomodulatory properties in the central nervous system [37]. It has been shown to regulate both the protein levels and mRNA expression of various cytokines, including TNF-α and IL-1β [38,39]. In the retina, our previous studies demonstrated that ouabain’s neuroprotective effects on RGCs rely on these cytokines [11]. Notably, ouabain rapidly increased the intracellular levels and release of IL-1β within 15 min of treatment [11]. In the present study, we observe a transient upregulation of active caspase-1, a key mediator in the maturation and release of IL-1β [22]. These rapid changes highlight ouabain’s ability to dynamically modulate IL-1β signaling, likely contributing to the early support of RGC survival. In addition, this aligns with previous findings sustaining a neuroprotective role of IL-1β to retinal cells [11].

Although IL-1β/IL-1R1 signaling has been implicated in cytotoxic processes during retinal degeneration [40], accumulating evidence points to its pro-survival role under certain conditions. Todd et al. [41] have demonstrated that IL-1β secreted by microglia interacts with IL-1R1 on astrocytes to confer neuroprotection of retinal cells exposed to N-methyl-D-aspartate [41]. This protective effect was diminished in IL-1R1-deficient mice but was restored when IL-1R1 expression was selectively reestablished in astrocytes. These findings suggest that astrocytic IL-1R1 is critical for microglia orchestrating inflammatory responses. Additionally, a recent study has shown that either a deficiency of IL-1R8 or excessive IL-1β can disrupt synapse morphology, plasticity, and function through the overactivation of IL-1R1 in mice. This dysregulation is linked to neurological impairments, highlighting the importance of balanced IL-1β/IL-1R1 signaling in maintaining neuronal health [42].

In this context, our findings reveal that ouabain dynamically modulates IL-1Ra levels to fine-tune IL-1β signaling, with IL-1Ra functioning as an essential counter-regulator of IL-1β activity [43]. The initial decrease in IL-1Ra at 15 min likely facilitates a transient period of enhanced IL-1R1 activation, possibly promoting an early neuroprotective response of IL-1β. This is followed by a compensatory increase in IL-1Ra at 48 h, which helps prevent prolonged IL-1β signaling and mitigates the risk of inflammatory damage. Importantly, low concentrations of IL-1Ra were sufficient to replicate the neuroprotective effects of ouabain and IL-1β on RGC, indicating its pivotal role in this signaling pathway. Notably, our results align with other in vitro data demonstrating that IL-1Ra treatment disrupts a caspase-1/IL-1β/IL-1R1 feedback loop induced by high glucose levels in human Müller cells, which drives sustained inflammatory signaling [44]. Our findings highlight a dual mechanism for ouabain, promoting IL-1β signaling for RGC survival while leveraging IL-1Ra to maintain signaling balance.

Using pharmacological inhibition assays, we identified Src, EGFR, and PKC as critical components of the signaling pathways mediating the neuroprotective effects of IL-1β and TNF-α. Specifically, IL-1β-induced RGC survival requires the activation of Src, EGFR, and PKC, while JNK activity is not essential. Interestingly, TNF-α also promotes RGC survival via EGFR and PKC, but this mechanism occurs independently of Src. In line with this, our previous findings demonstrated that PKC activation is essential for driving the release of IL-1β and TNF-α, ultimately enhancing RGC survival [45,46]. Collectively, our findings support a shared regulatory network linking ouabain, BDNF, IL-1β, and TNF-α through Src, EGFR, PKC, and JNK, suggesting that functional redundancy and crosstalk mechanisms enable cells to maintain functionality under stress conditions [47], thereby providing a protective advantage for RGCs following axotomy.

Our study has some limitations, particularly the exclusive use of retinal cell cultures, which, while offering controlled environments to investigate molecular and cellular mechanisms, do not fully capture the complexity of biological systems. Therefore, the findings presented here should be validated in vivo to better understand the complexity of these mechanisms, including cell–cell interactions within the retina. Additionally, relying solely on pharmacological approaches brings the risk of off-target effects, which could influence unrelated pathways and complicate the interpretation of our results. To enhance the robustness of our findings and provide a clearer understanding of how these signaling networks interact to promote RGC survival, incorporating genetic tools to manipulate these pathways would be a valuable complementary strategy. By integrating these approaches, future research will offer a more comprehensive understanding of the protective mechanisms involved, ultimately revealing novel targets for retinal neuroprotection and repair.

## Figures and Tables

**Figure 1 brainsci-15-00123-f001:**
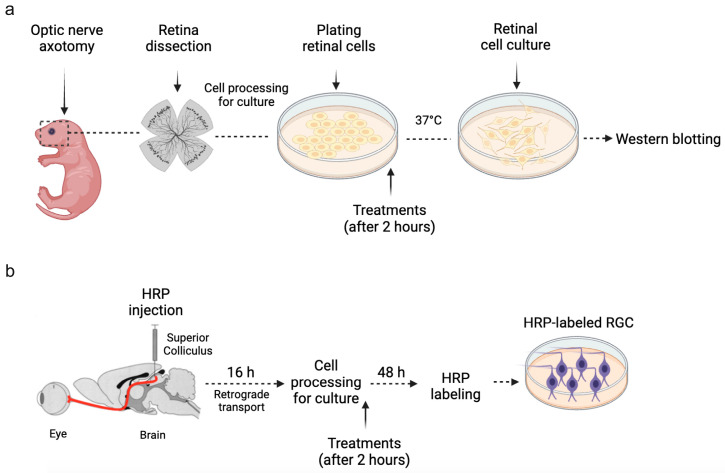
Experimental timeline. (**a**) Briefly, retinal cells from newborn rats were isolated, plated, and maintained in culture at 37 °C. Two hours after cell plating, retinal cell cultures were treated with ouabain and/or MMP-9 inhibitor. After different time intervals (15 min, 45 min, 24 h, or 48 h), the cells were processed for Western blotting. (**b**) For retrograde labeling of RGCs, horseradish peroxidase (HRP) was injected into each superior colliculus of newborn rats. After the time lapse necessary for retrograde transport of HRP to RGC soma (16 h), pups were euthanized and primary cultures from their retinas were prepared. Two hours after cell plating, retinal cell cultures were treated with IL-1β, IL-1Ra, or TNF-α. For mechanistic studies, retinal cultures were simultaneously exposed to selective inhibitors of signaling pathways. Forty-eight hours after the treatment, HRP-labeled RGCs were counted (Appendix A shows representative images of RGCs labelled with HRP).

**Figure 2 brainsci-15-00123-f002:**
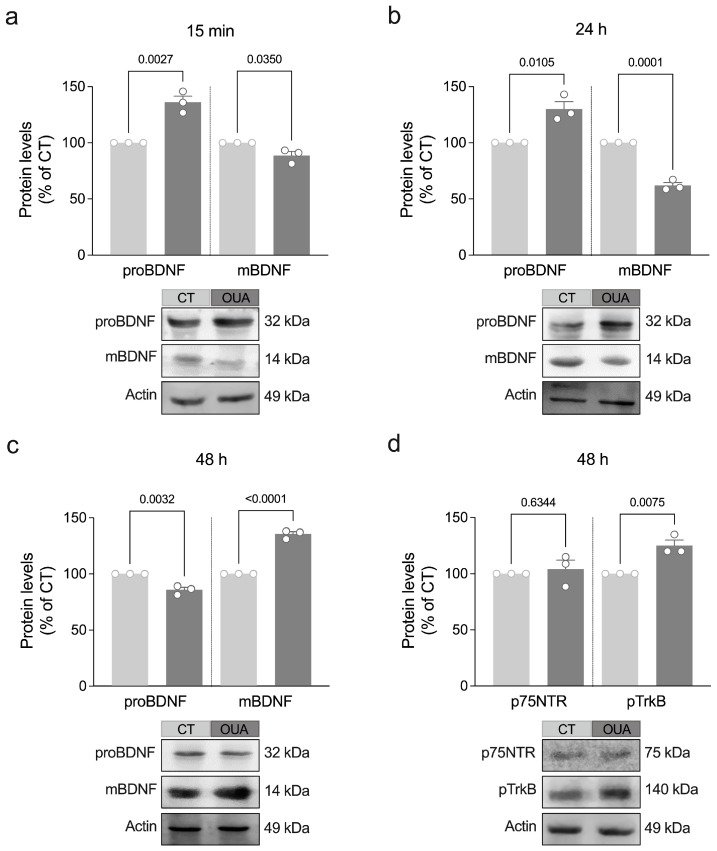
Ouabain regulates the maturation of BDNF and TrkB phosphorylation in retinal cells. (**a**–**d**) Primary retinal cultures were treated with ouabain (OUA; 3 nM) for the indicated periods for assessing levels of proBDNF, mBDNF, p75NTR, and pTrkB by Western blot. Plots represent mean (±SEM) protein optical density (normalized to actin) expressed as a percentage of control (CT) group (set at 100%), with N = 3 experiments using independent retinal cultures. Data were analyzed by a two-tailed Student’s *t*-test.

**Figure 3 brainsci-15-00123-f003:**
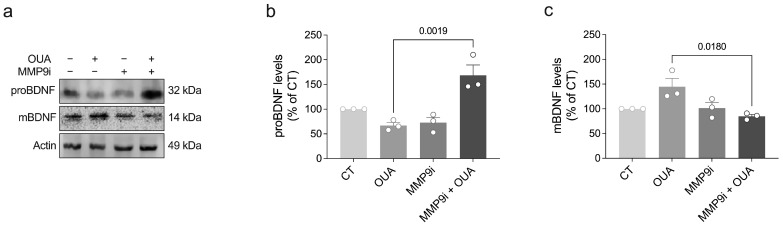
Inhibition of MMP9 activity prevents ouabain-induced maturation of BDNF in retinal cells. (**a**) Primary retinal cultures were treated with ouabain (OUA; 3 nM) and/or MMP9 inhibitor (MMP9i; 20 μM) for 48 h, and levels of (**b**) proBDNF and (**c**) mBDNF were assessed by Western blot. Plots represent mean (±SEM) protein optical density (normalized to actin) expressed as a percentage of control (CT) group (set at 100%), with N = 3 experiments using independent retinal cultures. Data were analyzed by one-way ANOVA followed by Holm–Sidak’s test.

**Figure 4 brainsci-15-00123-f004:**
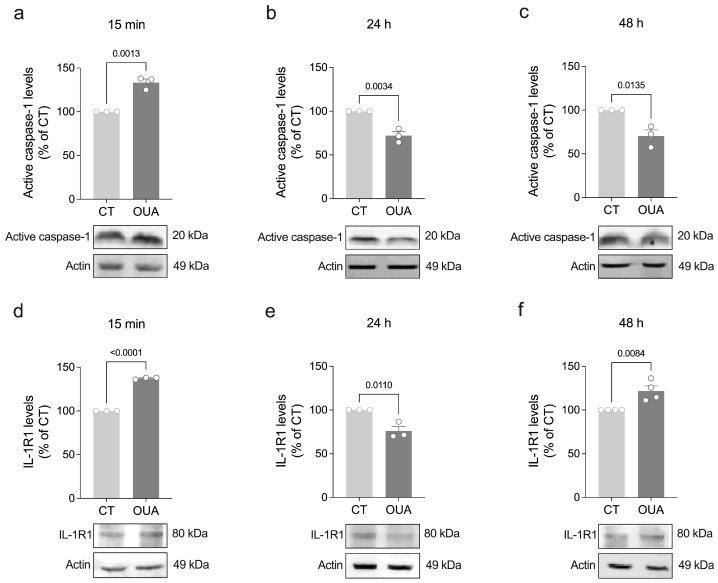
Ouabain regulates caspase-1 activity and IL-1R1 levels in retinal cells. Primary retinal cultures were treated with ouabain (OUA; 3 nM) for the indicated periods for assessing levels of (**a**–**c**) active caspase-1 and (**d**–**f**) IL-1R1 by Western blot. Plots represent mean (±SEM) protein optical density (normalized to actin) expressed as a percentage of control (CT) group (set at 100%), with N = 3–4 experiments using independent retinal cultures. Data were analyzed by two-tailed Student’s *t*-test.

**Figure 5 brainsci-15-00123-f005:**
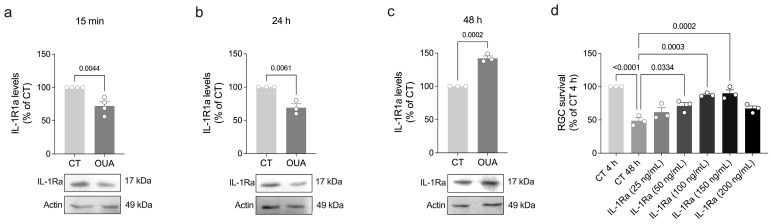
The IL-1R antagonist promotes RGC survival, and its levels are regulated by ouabain. (**a**–**c**) Primary retinal cultures were treated with ouabain (OUA; 3 nM) for the indicated periods for assessing levels of IL-1Ra by Western blot. Plots represent mean (±SEM) protein optical density (normalized to actin) expressed as a percentage of control (CT) group (set at 100%), with N = 3–4 experiments using independent retinal cultures. Additionally, (**d**) cultured retinal cells were exposed to IL-1Ra (25–200 ng/mL) for 48 h, and RGC survival was assessed by quantification of the number of HRP-labeled RGCs. Plots represent mean (±SEM) and are expressed as a percentage of control (CT) group at 4 h in culture (set at 100%), with N = 3 experiments using independent retinal cultures. Data were analyzed by (**a–c**) two-tailed Student’s *t*-test or (**d**) one-way ANOVA followed by Holm–Sidak’s test.

**Figure 6 brainsci-15-00123-f006:**
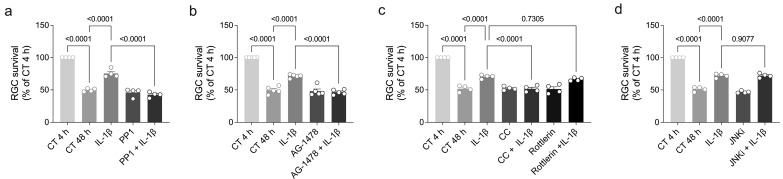
Src, EGFR, and PKC signaling pathways mediate IL-1β-induced RGC survival. Primary retinal cultures were treated with ouabain (OUA; 3nM) and/or the following drugs for 48 h: (**a**) PP1 (Src inhibitor; 1 μM), (**b**) AG-1478 (EGFR inhibitor; 2.5 μM), (**c**) chelerythrine chloride (CC; pan-PKC inhibitor; 1.25 μM), rottlerin (PKC-δ inhibitor; 2 μM), or (**d**) JNKi (JNK inhibitor; 0.5 μM). Plots represent mean (±SEM) RGC survival and are expressed as a percentage of control (CT) group at 4 h in culture (set at 100%), with N = 4–5 experiments using independent retinal cultures. Data were analyzed by one-way ANOVA followed by Holm–Sidak’s test.

**Table 1 brainsci-15-00123-t001:** Primary and secondary antibodies for Western blotting.

Name	Company	Dilution
** *Primary antibodies* **		
Rabbit anti-IL-1β	PeproTech (Cranbury, NJ, USA)	1:1500
Mouse anti-caspase-1	Santa Cruz Biotechnology (Dallas, TX, USA)	1:1000
Rabbit anti-IL-1Ra	PeproTech (Cranbury, NJ, USA)	1:1250
Mouse anti-IL-1R1	Santa Cruz Biotechnology (Dallas, TX, USA)	1:1000
Rabbit anti-mBDNF	PeproTech (Cranbury, NJ, USA)	1:1200
Mouse anti-proBDNF	Santa Cruz Biotechnology (Dallas, TX, USA)	1:1000
Mouse anti-p75NTR	Santa Cruz Biotechnology (Dallas, TX, USA)	1:1000
Rabbit anti-pTrkB	Santa Cruz Biotechnology (Dallas, TX, USA)	1:700
Rabbit anti-NGF	PeproTech (Cranbury, NJ, USA)	1:750
Rabbit anti-actin	Sigma-Aldrich (St. Louis, MO, USA)	1:500
** *Secondary antibodies* **		
Goat anti-rabbit IgG-HRP	GE Healthcare Life Sciences (Chicago, IL, USA)	1:15,000
Goat anti-mouse IgG-HRP	Santa Cruz Biotechnology (Dallas, TX, USA)	1:15,000

**Table 2 brainsci-15-00123-t002:** Signaling pathways underlying the trophic effect induced by ouabain, IL-1β, TNF-α, and BDNF on RGCs.

	Src	EGFR	PKC (PKC-δ)	JNK	References
Ouabain	Yes	Yes	Yes (PKC-δ-dependent)	Yes	[5,6]
IL-1β	Yes	Yes	Yes (PKC-δ-independent)	No	Figure 5
TNF-α	No	Yes	Yes (N/A)	N/A	Appendix A
BDNF	Yes	No	Yes (PKC-δ-dependent)	Yes	[7]

N/A (not assessed)**.**

## Data Availability

The original contributions presented in the study are included in the article and Appendix A. Further inquiries can be directed to the corresponding author.

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
