# Peer review of "Ouabain Counteracts Retinal Ganglion Cell Death Through Modulation of BDNF and IL-1 Signaling Pathways"

_brainsci, 2025, doi:10.3390/brainsci15020123_

Round 1
Reviewer 1 Report
Comments and Suggestions for Authors
The paper is interesting. However, some points to be addressed.
1 In this study, only in vitro cell experiments were performed, which appeared unconvincing in the exploration of the mechanism, and additional animal experiments were suggested to further elucidate them.
2 Statistical analysis:“First, all data were assessed for normal distribution by the Shapiro-Wilk test,then ANOVA or T test were examined”, how about the data that do not fit into a normal distribution?
3 Do the authors consider adding gene knock out or knock in treatment in cell experiments to further define the pathway?
Comments on the Quality of English Language
Quality of English Language is good.
Author Response
Point-by-point response
Reviewer #1
The paper is interesting. However, some points to be addressed.
- In this study, only in vitro cell experiments were performed, which appeared unconvincing in the exploration of the mechanism, and additional animal experiments were suggested to further elucidate them.
R: We thank Reviewer #1 for their interest and comments on our manuscript. Both in vitro and in vivo models have been widely used to study the mechanisms that regulate the survival of retinal ganglion cells (Yin & Benowitz, 2024). One major advantage of cell cultures is their ability to provide a higher level of experimental control compared to animal models, making them particularly relevant for mechanistic studies and the screening of trophic factors. However, we agree that animal experiments would be an interesting approach to corroborate the current in vitro data, as described in Discussion section (Page 11, lines 397-409). Currently, our laboratory has expertise in primary retinal cultures but not in in vivo models; therefore, animal experiments would take additional years (reagent orders, rat breeding, personnel training, pilot studies, experimentation itself) and would surpass by far the revision deadline for the current manuscript. We hope that this in vitro study will inspire interest from other research groups interested in RGC survival, enabling our findings to be tested and expanded using different approaches, including in vivo models.
- Statistical analysis:“First, all data were assessed for normal distribution by the Shapiro-Wilk test,then ANOVA or T test were examined”, how about the data that do not fit into a normal distribution?
R: We thank the reviewer for raising this point. In the present study, all data exhibited a normal distribution, as assessed by the Shapiro-Wilk test. Therefore, only parametric tests (ANOVA or Student’s t test) were used for the statistical comparisons. We modified the “2.6. Statistical analysis” section accordingly (Page 5, lines 168-169).
- Do the authors consider adding gene knock out or knock in treatment in cell experiments to further define the pathway?
R: Currently, we do not have access to genetic approaches to conduct such experiments, which would be highly valuable. However, we hope that publishing this manuscript will help us secure additional funding, allowing us to employ genetic approaches in the future to further investigate the cellular mechanisms through which ouabain prevents retinal ganglion cell death.
References
Yin Y, Benowitz LI. In Vitro and In Vivo Methods for Studying Retinal Ganglion Cell Survival and Optic Nerve Regeneration. InGlaucoma: Methods and Protocols 2024 Oct 22 (pp. 291-312). New York, NY: Springer US. https://doi.org/10.1007/978-1-0716-4140-8_22

Reviewer 2 Report
Comments and Suggestions for Authors
In this manuscript the authors explore the mechanisms by which oubain counteracts retinal ganglion cell death, using primary cultures of rat neural retina. I consider the manuscript well written, and the results are clearly aligned with the study's objectives. Although the results are encouraging, the manuscript could benefit from some adjustments, considering also that the Methods used are principally Western Blot analyses.
For better understanding of the work and how it is carried out, I suggest adding 2 images: a graphical abstract and an image in which is clearly shown the timeline followed for the retinal cells treatment. In this regard I also suggest implementing the Material and methods section adding a specific paragraph for the Cells Treatment. Despite they write all the drugs tested in the section 2.3, is not clear for how many times the cells have been treated.
Finally, to strengthen the data related to RGCs, it would be interesting to include representative images of the retrograde labeling.
I really appreciate the discussion: it is well written, clear describing also the limits of this work.
Author Response
Point-by-point response
Reviewer #2
In this manuscript the authors explore the mechanisms by which ouabain counteracts retinal ganglion cell death, using primary cultures of rat neural retina. I consider the manuscript well written, and the results are clearly aligned with the study's objectives. Although the results are encouraging, the manuscript could benefit from some adjustments, considering also that the Methods used are principally Western Blot analyses.
For better understanding of the work and how it is carried out, I suggest adding 2 images: a graphical abstract and an image in which is clearly shown the timeline followed for the retinal cells treatment. In this regard I also suggest implementing the Material and methods section adding a specific paragraph for the Cells Treatment. Despite they write all the drugs tested in the section 2.3, is not clear for how many times the cells have been treated.
Finally, to strengthen the data related to RGCs, it would be interesting to include representative images of the retrograde labeling.
I really appreciate the discussion: it is well written, clear describing also the limits of this work.
R: We are grateful to the reviewer for their comments and suggestions. First, we created a graphical abstract for the manuscript (please see below). We also made some modifications to the “Materials and Methods” section (page 3, lines 109-114; lines 119-129; page 4, line 133) and added a figure to clarify the experimental design, as suggested (New Figure 1). We emphasize that all cultures were treated only once, two hours after cell plating, regardless of the experiment's duration. Additionally, representative images of RGC labeling have been included in the Supplementary Material (New Figure S1; page 4, line 149). We sincerely appreciate the suggestions, which undoubtedly improved the quality of our manuscript.
